# Asymptomatic Autonomic Dysregulation after Recovery from Mild COVID-19 Infection Revealed by Analysis of Heart Rate Variability Responses to Task Load

**DOI:** 10.3390/healthcare12010043

**Published:** 2023-12-24

**Authors:** Toshikazu Shinba, Yujiro Shinba, Shuntaro Shinba

**Affiliations:** 1Department of Psychiatry, Shizuoka Saiseikai General Hospital, Shizuoka 422-8527, Japan; 2Research Division, Saiseikai Research Institute of Health Care and Welfare, Tokyo 108-0073, Japan; 3Autonomic Nervous System Consulting, Shizuoka 420-0839, Japan

**Keywords:** post-COVID-19, asymptomatic autonomic dysregulation, heart rate variability, task load

## Abstract

(1) Background: The coronavirus disease 2019 (COVID-19) infection is often followed by various complications, which can cause disturbances in daily life after recovery from the infectious state, although etiological mechanisms are not fully elucidated. Previous studies have indicated that autonomic dysregulation is an underlying factor, and it is of interest to clarify whether autonomic dysregulation is also present in the asymptomatic subjects after COVID-19 infection (post-COVID-19) for early detection of post-COVID-19 complications. (2) Methods: In the present study, autonomic activity was assessed using heart rate variability (HRV) analysis in the workers who recovered from mild COVID-19 infection (*n* = 39). They took a leave of absence for an average of 11.9 days and returned to the original work without complications. HRV was measured after an average of 9.3 days from return. High-frequency (HF) and low-frequency (LF) HRV parameters and heart rate (HR) were recorded during a three-behavioral-state paradigm of approximately 5 min length composed of initial rest, task load, and post-task rest periods and were compared with the data of the workers without the history of COVID-19 infection (normal, *n* = 38). (3) Results: The HRV and HR scores at the initial rest in the post-COVID-19 subjects showed no difference from those in the control. It is found that the post-COVID-19 subjects exhibited an attenuation of LF/HF increment during the task load and an excessive increase of HF together with a decrease of LF, LF/HF and HR during the post-task rest period in comparison with the initial rest scores. (4) Conclusions: These abnormalities are evaluated as asymptomatic autonomic dysregulation in response to task load, are frequently present after COVID-19 infection, and could be related to the generation of complications.

## 1. Introduction

The coronavirus disease 2019 (COVID-19) is a systemic disorder affecting not only the respiratory but various systems, producing a wide range of symptoms including cough, dyspnea, anosmia, dysgeusia, memory problem, brain fog, sleep disturbance, anxiety, depression, palpitation, and arrhythmia [1,2,3]. The symptoms also cover general somatic manifestations, such as fatigue and pain. These symptoms are found not only during the COVID-19 acute phase but also in the post-infectious period [1]. The prevalence rate is 20 to 40%, and the symptoms generate serious problems for the patient’s life after the recovery from acute infection [4]. In order to deal with these problems, evidence-based clinical communication is important including the etiological mechanisms [5].

The mechanisms of generation underlying these varieties of symptoms are not fully clarified. Previous works support that autonomic dysregulation can be one of the factors involved in COVID-19 complications, and heart rate variability (HRV) has been used for the analyses [6]. It was reported that immune function, D-dimer, and NT-proBNP were related to HRV indices in COVID-19 patients with severe symptoms, and their relation is linked to poor prognosis [7]. A study of sudden cardiac death in COVID-19 patients indicated the presence of low parasympathetic activity shown by HRV analysis preceded the cardiac disturbances [8]. Sleep architecture alteration by COVID-19 infection was also found using HRV measurements [9].

Disturbed conditions after recovery from the acute infectious processes were also analyzed using HRV. HRV derangement was related to the severity of cardiovascular complications following COVID-19 infection [10]. Postural orthostatic tachycardia syndrome, orthostatic hypotension, and inappropriate sinus tachycardia were also accompanied by a decrease in HRV parameters [11,12].

In dealing with these COVID-19 complications accompanying autonomic dysregulations, autonomic analyses in the asymptomatic subjects would be important for early detection of and adequate intervention to the post-COVID-19 disturbances. Preclinical autonomic conditions in post-COVID-19 patients have been checked in several studies using 24 h Holter ECG recordings. In a study on post-COVID-19 patients, both time-domain and frequency-domain HRV indices were high [13]. Other studies showed that HRV indices were low [14,15]. The results using 24 h Holter ECG are not consistent probably because the data were collected during various behavioral states, sleep, work, meal, etc., which could differentially affect the HRV parameters. The control of behavioral state during HRV recordings will be important for analyzing the autonomic activities.

HRV is the fluctuation of inter-heartbeat intervals and is known to modulate blood flow by changing heartbeats in response to alterations of various psychosomatic conditions in relation to respiration and blood pressure [16,17]. Adequate HRV leads to a stable blood supply by adjusting the heart rate. Power spectrum analysis reveals mainly two types of fluctuation, high-frequency (HF) and low-frequency (LF) components; frequency bands are 0.15–0.4 Hz and 0.04–0.15 Hz, respectively [16]. HF component reflects parasympathetic activity [18], and LF is closely related to blood pressure Mayer wave, reflecting both sympathetic and parasympathetic activities in relation to baroreflex function [17]. HF, LF, and their ratio (LF/HF) are often utilized to analyze the sympathetic and parasympathetic activities of various somatic and mental disorders, including ischemic heart disease, depression, and anxiety [19,20].

Based on the above-stated issues, the present study aimed to control the behavioral state by measuring HRV in the well-controlled behavioral paradigm composed of three behavioral states [20]; initial resting (Rest), task performing (Task), and post-task resting (After) states as described below, and applied this paradigm to measure HRV in asymptomatic post-COVID-19 subjects. The three-behavioral-state paradigm has been used in our previous studies on depression, anxiety, and fatigue to reveal HRV disturbances [20]. In healthy subjects, HF reacts to the task load by decrement and LF/HF by increment. After the task, these changes return to the normal level. In depression, anxiety, and fatigue, not only the Rest data but the Task and After data were altered. It is of interest to apply this paradigm to post-COVID-19 subjects to verify the presence of autonomic derangement reflected in HRV. As for the task load, the random number generation task has been continuously employed in our studies because it is simple and can be performed by subjects with different psychological, educational, and intellectual conditions in daily clinical practices [21] and also employed in the present study. The comparison of the present data with our previous data including the normal control is also possible by using this task.

If a certain pattern of autonomic dysfunction is found in the asymptomatic state after COVID-19 infection, it may be used as a warning sign for the occurrence of complications and will be important information in the health management of COVID-19 infection.

## 2. Materials and Methods

### 2.1. Subjects

Thirty-nine workers who recovered from mild COVID-19 infection voluntarily participated in the present study between April 2022 and July 2022 (age, 33.3 ± 11.2 years, mean ± S.D.; 8 males and 31 females, post-COVID-19). They were diagnosed with COVID-19 by polymerase chain reaction (PCR) test at Shizuoka Saiseikai General Hospital, exhibited mild symptoms with no need for treatment in the hospital, and stayed home during the infectious period. They were medical staff at the hospital and their PCR tests were found positive in their medical check. All of them had received COVID-19 vaccination more than a month before the infection. The prevalence of symptoms during their infectious periods is presented in Table 1. No complication was noted at the time of their HRV measurements. They were off from work for 11.9 ± 2.4 days after the PCR test was found positive and returned to work without complications. HRV was recorded at 21.2 ± 7.1 days after the PCR test, at 9.3 days from the return on average. All subjects at the time of HRV measurement conducted their original work. The subjects were excluded from the present study when their symptoms got severe and needed to be treated.

The HRV data of the post-COVID-19 subjects were compared with the data of the age and gender-matched normal controls (*n* = 38, age, 36.7 ± 9.5 years; 12 males and 26 females) who never got COVID-19 infection. They were not medical staff, and their data were collected before the outbreak of COVID-19 infection in 2019 in our continuous HRV study project. Enrollment of medical staff for the normal control subjects was not adequate at the time of the pandemic because the use of PCR was limited to symptomatic subjects, and the inclusion of subjects with asymptomatic infection was unavoidable.

The three-behavioral-state paradigm described below was conducted by the post-COVID-19 subjects and by the normal control subjects in the completely same experimental setting to enable the comparison of the data between the two groups. Psychophysical conditions including depressiveness, anxiety, stress, and fatigue were also checked by two questionnaires, State and Trait Anxiety Inventory state (STAI-state) score and Self-rating Depression Scale (SDS) score [22,23] in both the post-COVID-19 subjects and the normal control. It has been reported that medical staff is under stressful working conditions, and a comparison between the data of the medical staff and that of the non-medical staff should be carried out carefully [24]. However, SDS score and STAI-state score were 39.4 ± 8.0 and 42.4 ± 10.9 in the post-COVID-19 subjects and were not statistically different from those in the normal control (SDS, 40.9 ± 7.9, *p* > 0.05; STAI-state; 40.3 ± 8.4, *p* > 0.05). It was assumed that the psychophysical condition of the subjects was not the major factor affecting the present results.

The age and male-to-female ratio showed no statistically significant differences between the post-COVID-19 and the normal control subjects (age, difference = −4.0, 95% CI = −8.0 to 0.0, *p* > 0.05; male-to-female ratio, relative risk = 0.861, 95% CI = 0.641 to 1.128, *p* > 0.05). All subjects had no history of neurological, endocrinological, cardiological, arrhythmic, or psychiatric disorders and were not taking medications at the time of the HRV measurement. The subject did not take caffeine for at least an hour before the measurement. Smokers and daytime alcohol drinkers were not enrolled.

The present study was performed in accordance with the Declaration of Helsinki. Written informed consent to participate in this study was obtained from all subjects. The protocol of the study was approved by the institutional review board of Shizuoka Saiseikai General Hospital (No 24-10-03).

### 2.2. Heart Rate Variability Measurement

The protocol of heart rate variability measurement is the same as used in our previous studies, and the details are found in our publication [20]. The measurement was conducted between 14:00 and 17:00 while the subject was seated on a chair after 5 min of adaptation without physical activity. A wearable electrocardiogram (ECG) device was put on the chest (RF-ECG2, GM3, Tokyo, Japan), and the recorded ECG was stored in a computer. The R-R interval fluctuations were analyzed using the maximum entropy method (MemCalc, GMS, Tokyo, Japan). There are several methods for the power spectrum analysis of R-R trend data. We used the maximum entropy method because it has been successfully applied to trend data with a minimum duration of 30 s [25]. The Fast-Fourier Transform (FFT) method has been frequently employed for power spectrum analysis but usually requires at least 256 data, which corresponds to about 5 min in the case of the R-R trend. The task load of 5 min duration can cause distress depending on the subject, and the control of steady task performance would be difficult. MemCalc [26] can analyze the data of short sample series and is adequate for the three-behavioral-state paradigm in the present study, with each behavioral state of about 1 min length [20,25]. The scores using MemCalc and that using FFT are different, and it was necessary to employ the same power spectrum analysis method throughout our project.

The present ECG device (RF-ECG2, GM3, Tokyo, Japan) is small and wearable to minimize the distress of the subjects and sampled ECG signals at the frequency of 200 Hz. This sampling rate can cause an inaccurate estimation of the R-R interval up to 5 ms, which is 0.5% of the R-R interval data of a subject with a heart rate of 60/min. After peak detection, R-R intervals between the range of 273 and 1500 ms were used for power spectral analysis to exclude paroxysmal beats. When an R-R interval was omitted, it was replaced by the average of the preceding and following intervals. These R-R intervals were resampled at the mean HR. This ECG sampling method has been successfully utilized in the previous studies [20].

Based on the standards of the European Society of Cardiology and the North American Society of Pacing and Electrophysiology [16], MemCalc calculated low-frequency (LF) and high-frequency (HF) components of the spectrum every 2 s by integrating the power at frequency intervals of 0.04–0.15 Hz for LF and of 0.15–0.4 Hz for HF during the preceding 30 s period. HR (/min) was calculated from the R-R intervals. It is known that HF reflects parasympathetic activity related to breathing frequency [18]. Breathing activity was monitored by the experimenter, and its frequency was confirmed to be within the range of 0.15-0.4 Hz in each subject, as previously indicated [27]. When the breathing frequency was found to be out of this frequency range, the subjects were asked to modulate the breathing, and the measurement was restarted from the beginning.

ECG was recorded in three different states. First, in the initial resting state (Rest), the subjects were relaxed in the chair for approximately 60 s. Then, during the task load (Task), the subjects performed a random number generation task for 100 s. After the task (After), ECG was measured for 60 s period in a relaxed condition. LF, HF, LF/HF, and HR scores were averaged in the interval from 30 s after the onset to the end of each condition to exclude any data at the beginning of each new period that can reflect the previous state (AMAS, GM3, Tokyo, Japan). The flow of the paradigm is listed below.
Rest state 60 sTask state (Random number generation task) 100 sPost-task rest state 60 s

In the random number generation task, the subjects generated orally the numbers 0 through 9 at a random order 100 times at the rate of 1 Hz. The rate was indicated by a click sound. The subjects were asked to concentrate on this task, and all completed the task. Randomness in the generated numbers was evaluated using counting bias (CB; frequency of counting up or down), interval bias (IB; frequency of same interdigit intervals), and random number generation index (RNG; frequency of same digit pairs) to check the task performance. Details were found in the previous research [20,21].

### 2.3. Statistics

The differences in the HRV indices at the Rest, Task, and After periods in each subject group were checked by repeated measure analysis of variance (ANOVA) with post hoc Tukey’s multiple comparison test. The differences in the HRV and HR indices in each period between the post-COVID-19 and normal control subjects were examined by the Mann–Whitney U-test, because some of the HRV and HR data did not show normal distribution. The differences in the task performance indices (CB, IB, and RNG), the age, and the STAI-state and SDS scores between the post-COVID-19 and normal control subjects were also checked by the Mann–Whitney U-test. The male-to-female ratio of the subjects was examined by Fisher’s exact test. 95% confidence intervals as well as the mean differences were shown as statistical descriptions (Prism 8, GraphPad Software, San Diego, CA, USA). The sample size was calculated with G-Power (ver. 3.1, α error probability = 0.05, 1-β error probability = 0.95) when the statistical significance was found between the post-COVID-19 and the normal control subjects (G*Power Version 3.1.9.6, Franz Faul, Universität Kiel, Kiel, Germany).

## 3. Results

The HRV and HR data during the Rest, Task, and After states together with the ratios (Task/Rest, After/Rest) were presented in Table 2. Figure 1 shows the HRV and HR profiles during the three-behavioral-state paradigm. Both for the normal control and the post-COVID-19 data, ANOVA revealed a significant effect of behavioral state on HF, LF/HF, and HR, and the F (2,76) scores are 26.7, 19.9, and 46.8 in the normal control and 29.7, 13.2, and 40.0 in the post-COVID-19 data (*p* < 0.05), respectively.

Figure 2 presents the ratios of the HRV and HR indices during the Task state to those during the Rest state (Task/Rest) and to those during the After state to those during the Rest state (After/Rest). The data were exhibited as box-whisker data in Tukey’s format. Differences in the Task/Rest and After/Rest ratios for each HRV and HR indices between the groups were checked with the Mann–Whitney U-test, and a significant difference was shown by an asterisk.

### 3.1. LF

The scores during each behavioral state (Rest, Task, After) did not show group differences (Figure 1, LF, *p* > 0.05). It was also found that LF did not show consistent responses to task load in both normal control and post-COVID-19 subjects (Figure 1, LF, *p* > 0.05). However, the After/Rest ratio was significantly lower in post-COVID-19 subjects (Figure 2, LF, U = 513.5, *p* = 0.02, sample size = 1121). The Task/Rest ratio showed no group difference (Figure 2, LF, *p* > 0.05).

### 3.2. HF

The HF scores at Rest, Task, and After states in the post-COVID-19 subjects showed no differences in comparison with those in the normal control subjects (Figure 1, HF, *p* > 0.05). Repeated measure ANOVA indicated that the HF score at the Task state showed a reduction from the Rest score both in the normal control (Figure 1, HF, T, difference = 292.4, 95% CI = 168.0 to 416.9, *p* < 0.0001) and in the post-COVID-19 subjects (difference = 136.5, 95% CI = 32.1 to 240.9, *p* = 0.0079). The HF score at the After state showed an increment from the Rest score only in the post-COVID-19 subjects (Figure 1, HF, A, post-COVID-19, difference = −364.5, 95% CI = −548.3 to -180.6, *p* < 0.0001). In the normal control subjects, the HF score at the After state was not different from the Rest score (Figure 1, HF, A, normal, difference = −43.8, 95% CI = −131.8 to 44.2, *p* > 0.05). The reduction rate in the post-COVID-19 subjects was also not different from that in the normal control subjects (Figure 2, HF, Task/Rest, *p* > 0.05). On the other hand, the After/Rest ratio was significantly greater in the post-COVID-19 subjects than that in the control (Figure 2, HF, U = 390.5, *p* = 0.0003, sample size = 31).

### 3.3. LF/HF

The LF/HF scores at the Rest and Task states in the post-COVID-19 subjects were not different from those in the normal control subjects (*p* > 0.05). But the scores at the After state in the post-COVID-19 subjects were significantly lower than that in the normal control subjects (U = 401, *p* = 0.0004, sample size = 34). Repeated ANOVA indicated that in the normal control subjects, LF/HF score increased during the Task state (Figure 1, LF/HF, T, normal, difference = −1.82, 95% CI = −2.62 to −1.01, *p* < 0.0001) and returned to the baseline Rest level at the After state (Figure 1, LF/HF, A, normal, difference = −0.18, 95%CI = −0.84 to 0.48, *p* > 0.05). In the post-COVID-19 subjects, LF/HF score at the Task state was not different from that in the Rest state (difference = −0.22, 95% CI = −1.33 to 0.88, *p* > 0.05), but LF/HF score at the After state was significantly lower than that in the Rest state (Figure 1, LF/HF, A, post-COVID-19, difference = 1.76, 95% CI = 0.56 to 2.9, *p* = 0.0025). The After/Rest ratio of LF/HF in the post-COVID-19 subjects was also smaller than that in the normal control subjects (Figure 2, LF/HF, After/Rest, U = 308, *p* < 0.0001, sample size = 100).

### 3.4. HR

HR scores at Rest, Task, and After states in the post-COVID-19 subjects were not different from those in the normal control subjects (Figure 1, HR, R, *p* > 0.05). In both normal control and post-COVID-19 subjects, HR increased significantly during the Task state (difference = −9.7, 95% CI = −13.1 to −6.2, *p* < 0.0001 in normal control; difference = −6.6, 95% CI = −9.7 to −3.5, *p* < 0.0001 in the post-COVID-19 subjects). HR returned to the Rest level during the After state (difference = 0.5, 95% CI = −0.5 to 1.4, *p* > 0.05) in the normal control subjects with the averaged After/Rest score being 0.99. In the post-COVID-19 subjects, HR at the After state was lower than that at the Rest (difference = 2.9, 95% CI = 1.5 to 4.3, *p* < 0.0001), and the After/Rest ratio in the post-COVID-19 subjects was lower than that in the normal control subjects (Figure 2, U = 417.5, *p* = 0.0008, sample size = 43).

### 3.5. Task Performance

As for the task performance of random number generation, CB, IB, and RNG scores in the post-COVID-19 subjects are 0.163 ± 0.083, 0.606 ± 0.141, and 0.340 ± 0.043, respectively, and are not different from those in the normal control subjects (CB: 0.130 ± 0.069, IB: 0.569 ± 0.109, RNG: 0.340 ± 0.058, *p* > 0.05).

## 4. Discussion

The present study examined the HRV and HR profiles during the three-behavioral-state paradigm consisting of Rest, Task, and After states in the asymptomatic post-COVID-19 subjects and found both similarities and differences in comparison with the normal control subjects. The task performance as evaluated with CB, IB, and RNG indices in the post-COVID-19 subjects was not different from that in the normal control subjects. It is indicated that the task performance did not significantly affect the present findings.

At the baseline Rest state, all the HRV indices including LF, HF, and LF/HF, as well as HR, in the asymptomatic post-COVID-19 subjects are not different from those in the normal control subjects (Figure 1). It was suggested that the autonomic activity reflected in HRV and HR is normally regulated when no task or stress is assigned to the post-COVID-19 subjects. A previous study on fatigued post-COVID-19 subjects also showed no change in the baseline autonomic activity examining HR and blood pressure [28], supporting that autonomic activity is not affected at the resting condition when the patients show no severe symptoms.

The use of task load in the three-behavioral-state paradigm detected HRV abnormalities in the present study. During the task, an increment of LF/HF score, which was normally present, was attenuated in the post-COVID-19 subjects (Figure 1, LF/HF; Figure 2, LF/HF, Task/Rest). Other HRV and HR indices during the Task state showed no abnormalities in the post-COVID-19 subjects. It has been reported that the subjects with depression and chronic fatigue syndrome exhibit a significant attenuation of HF reduction during the Task state. This modulation of autonomic function during the task load was found intact in the post-COVID-19 subjects, suggesting that the pathophysiological condition in the asymptomatic post-COVID-19 autonomic dysregulation is different from that in depression and chronic fatigue syndrome. However, the attenuated response of LF/HF during the task load, found in the post-COVID-19 subjects, is present in depression and chronic fatigue syndrome. It could be important to assess the possibility that the asymptomatic post-COVID-19 subjects may develop depression or chronic fatigue syndrome.

The After state was the major condition when autonomic dysregulation occurred in the post-COVID-19 subjects. The present study revealed significant changes in all HRV and HR indices in the post-COVID-19 subjects at the After state (Figure 1 and Figure 2). An increase in HF, together with a decrement in LF, LF/HF, and HR, was noted in comparison with the Rest score when the task was unloaded. The three-behavioral-state paradigm is effective in detecting these autonomic abnormalities after the task load. The functional significance of the results during the After state can be evaluated as concomitant parasympathetic hyperactivation and sympathetic deactivation after the task is over, reflected in the HF increment and LF decrement surpassing the initial Rest scores. These observations imply that the autonomic balance, sympathetic vs. parasympathetic, is shifted toward parasympathetic activation after the execution of tasks. However, the sample size of the After/Rest ratio of LF data was high, suggesting that the results on LF would not be as valid as those on HF, LF/HF, and HR exhibiting smaller sample sizes. Future studies are warranted to examine the relationship between this autonomic dysregulation after the task load and various post-COVID-19 complications, including depression and fatigue.

The present study has limitations. The normal control subjects were not the medical staff to avoid the inclusion of the subjects with subclinical COVID-19 infection. The use of medical staff as the control was not adequate at the time of the pandemic as described in the Materials and Methods section. In spite of the difference in the work between the post-COVID-19 subjects and the normal control subjects in the present study, psychophysical conditions evaluated by questionnaires showed no differences between the two groups, indicating that the workplace would not be the major factor contributing to the present results. However, future studies on subjects with the same working environment should be important. The limitations also include the lack of analysis on the menstrual phase in the female subjects [29], on the circadian changes of HRV [30], and on the content of meals during the experimental day. These factors have been reported to influence the baseline HRV and HR scores. In spite of these limitations, the present study revealed that the HRV and HR scores at the Rest state were not altered in the post-COVID-19 subjects. If the stressful environment at work, the hormonal influence, and the nutritional differences had significant effects on the results, the Rest score would possibly be altered. In previous studies on depression and chronic fatigue syndrome, for which stress often acts as a major precipitating factor, the Rest score was found to be significantly low [20]. The absence of alterations at the Rest score for all HRV and HR indices in the post-COVID-19 subjects suggests that the present finding on the HRV and HR scores at the Task and After states may reflect the influence of COVID-19 infection. Future studies are interesting to examine the effects of stress, hormonal changes, circadian rhythm, and nutritional conditions on the HRV and HR response to task load in the present three-behavioral-state paradigm. The use of a higher sampling frequency and a longer period for each behavioral state would also be necessary to consolidate the present findings.

The functional significance of these HRV and HR abnormalities can be assessed as aberrant responses of autonomic activity to stress. Autonomic derangement in asymptomatic post-COVID-19 subjects is not salient at the resting state and during the task load but becomes apparent after the task is over and stress is reduced. This HRV profile in the post-COVID-19 subject is different from that in major depressive disorder, chronic fatigue syndrome, and normal subjects with elevated depressiveness and anxiety who show HRV abnormalities at the Rest and Task states [20]. It has been reported that resting HRV can serve as a putative biomarker of stress resilience. Higher baseline resting HRV is related to emotion regulation [31]. An association between HRV and worries was also indicated in post-COVID-19 subjects, supporting that high HRV can be protective against stress [32]. High HF at the After state in the post-COVID-19 subjects in the present study may be related to the increased resilience reducing the effects of stress. The existence of HRV changes without somatic abnormalities can indicate the presence of silent autonomic alteration and may be used for early detection of post-COVID-19 effects [1,2,3]. Future studies are warranted to verify the clinical course of asymptomatic post-COVID-19 subjects showing HRV derangements.

## 5. Conclusions

The present study has indicated that the HRV measurement during the three-behavioral-state paradigm consisting of the Rest, Task, and After states can reveal autonomic dysregulation in asymptomatic post-COVID-19 subjects. The autonomic dysregulation is mainly observed after the task is unloaded and could be related to the pathophysiology of post-COVID-19 conditions. The existence of HRV changes without somatic abnormalities can indicate the presence of silent autonomic alteration and may be used for early detection of complications.

## Figures and Tables

**Figure 1 healthcare-12-00043-f001:**
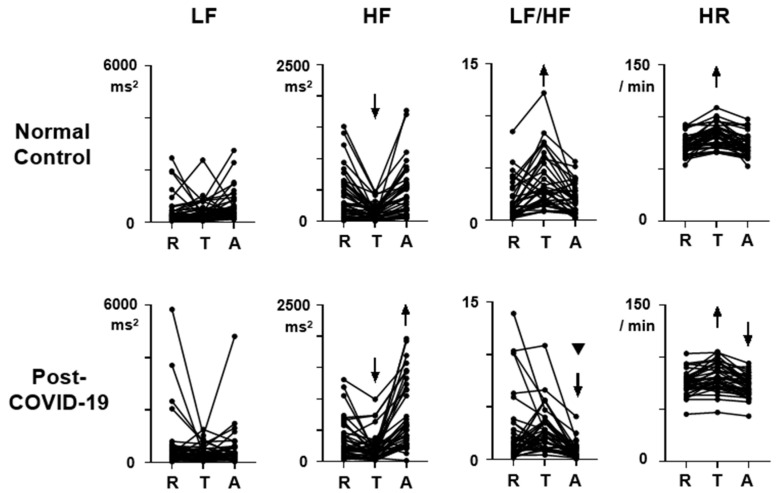
Heart rate variability (HRV) indices and heart rate (HR) profiles during the initial rest (R), the random generation task (T), and the rest-after-task (A) conditions in the normal control and post-COVID-19 subjects. HRV indices include low-frequency component (LF), high-frequency component (HF), and their ratio (LF/HF). Each filled circle indicates individual data, and the data from the same individuals are connected by lines. Significant increases and decreases from the score during the rest (R) are represented by upward and downward arrows, respectively (*p* < 0.05, Tukey multiple comparison test), above the data during the task (T) and the rest after the task (A). The filled reversed triangle indicates that the data of post-COVID-19 in the behavioral state are significantly lower than that of the normal control analyzed with the Mann–Whitney U-test (*p* < 0.05).

**Figure 2 healthcare-12-00043-f002:**
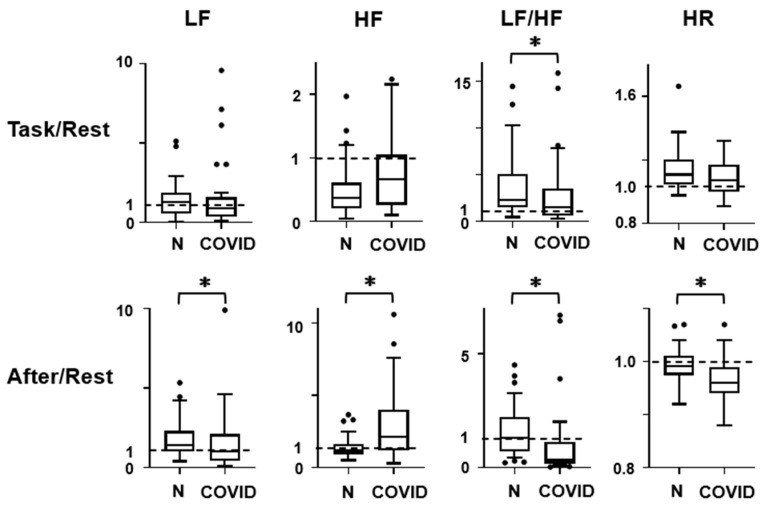
The ratio of the data during the task to that during the rest (Task/Rest) and the ratio of the data during the rest after the task to that during the rest (After/Rest). The ratios were compared between the normal control (N) and post-COVID-19 (COVID) subjects for HRV indices, low frequency (LF), high frequency (HF), and their ratio (LF/HF) together with heart rate (HR). The data were presented in box/whisker format of Tukey’s format. A significant difference between the data of normal control and post-COVID-19 subjects was indicated by an asterisk (Mann–Whitney U-test, *p* < 0.05).

**Table 1 healthcare-12-00043-t001:** The prevalence of symptoms during the infectious period in the post-COVID-19 subjects.

Fever	52.8%
Headache	47.2%
Sore throat	58.3%
Pain	30.6%
Dyspnea	5.6%
Cough	75.0%
Dysgeusia	8.3%
Anosmia	13.9%

**Table 2 healthcare-12-00043-t002:** HRV and HR data in the three-behavioral-state paradigm.

		Rest	Task	After	Task/Rest	After/Rest
LF	N	449.5 ± 553.1	418.0 ± 420.1	605.8 ± 585.3	1.42 ± 1.14	1.83 ± 1.13
(ms^2^)	COV	656.5 ± 1089.1	377.7 ± 277.4,	515.1 ± 775.5	1.51 ± 1.97	1.55 ± 1.76
HF	N	445.1 ± 380.4	152.7 ± 125.6	488.9 ± 411.7	0.50 ± 0.41	1.33 ± 0.69
(ms^2^)	COV	345.2 ± 312.7	208.7 ± 218.3	709.7 ± 541.0	0.77 ± 0.58	2.96 ± 2.33
LF/HF	N	1.69 ± 1.81	3.50 ± 2.56	1.87 ± 1.45	3.88 ± 3.46	1.60 ± 1.07
	COV	2.59 ± 3.25	2.81 ± 2.07	0.83 ± 0.75	2.84 ± 3.48	0.93 ± 1.49
HR	N	72.8 ± 9.1	82.5 ± 10.4	72.4 ± 9.7	1.14 ± 0.14	0.99 ± 0.03
(/min)	COV	74.8 ± 10.7	81.4 ± 12.6	72.0 ± 10.2	1.09 ± 0.10	0.96 ± 0.04

The data are presented as mean ± S.D. The units of the data in Rest, Task, and After states were present in the left column. N, normal control; COV, post-COVID-19.

## Data Availability

The data that support the findings of this study are available on request from the corresponding author. The data are not publicly available due to privacy and ethical restrictions.

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
