# Peer review of "Asymptomatic Autonomic Dysregulation after Recovery from Mild COVID-19 Infection Revealed by Analysis of Heart Rate Variability Responses to Task Load"

_healthcare, 2023, doi:10.3390/healthcare12010043_

Round 1

Reviewer 1 Report

Comments and Suggestions for Authors

Please note the appendix. 

Comments on the Quality of English Language

As a non-native English speaker, it seems adequate. 

Author Response

Dear Reviewer 1

 Thank you for all of the important comments and suggestions. We modified the manuscript following them, and hope that the manuscript is improved enough. Below is the list of your comments together with our responses.

Sincerely yours,

Toshikazu Shinba

Department of Psychiatry

Shizuoka Saiseikai General Hospital

General Comment:

While reading the paper, some thoughts occurred to me, which I would now like to present to you. I am a supporter of HRV methodology, but I see major weaknesses here in the methodology and in the consideration of confounders. I find the major problem in the comparison of two study groups, between whose studies there are several years and for me there is no comparability, especially since study conditions were not shown.

Response:

 Thank you for the important and valuable comments. As for the major point of your comment regarding the comparability of post-COVID19 subjects and the normal control subjects, we described the details in the Subjects subsection (Materials and Method, line 105, line 116). We used the normal controls subjects before the outbreak of COVID-19 infection because it was not possible to enroll the subjects at the time of pandemic completely free from the COVID-19 infection with limited PCR supply. The psychophysical conditions were checked in both groups using questionnaires and their scores in two groups are not different. The paradigm and methods we employed in the present study were also completely same as those used for the normal control subjects. Based on these backgrounds, we compared the HRV and HR data in the post-COVID-19 subjects and those in the control subjects. We also described the limitations of the present finding in the Discussion section (line 360). The details are also found below.

Comment1) Please place the objective of the study at the end of the introduction. What hypotheses did you follow?

Response: Thank you for the valuable comment. Following your suggestion, the objective and hypotheses of the present study were moved to the end of the introduction, stating ‘If a certain pattern of autonomic dysfunction is found in the asymptomatic state after COVID-19 infection, it may be used as a warning sign for the occurrence of complication and will be an important information in the health management of COVID-19 infection.’ (line 95).

2) Lines 48-50: Here you should rephrase and add data. How can you measure a reduced HRV during an autopsy?

Response: Thank you for the question and comment. The sentence regarding the referenced article was rephrased (line 50).

3) Lines 79-91: LF reflects parasympathetic and sympathetic activity. Please add.

Response: Thank you for the comment. Following your comment, a sentence was added to describe that LF reflects both sympathetic and parasympathetic activities (line 75).

4) The methods of task performance are not discussed at all in the introduction. Why was this surveyed at all? Indicate the current state of the science relating to task performance (applied methods) and Covid-19 infection.

Response: Thank you for the comment. We explained the reason to use random number generation task at the introduction section. This task has been utilized in various studies including our previous ones. It can be performed by the subjects of different psychological, educational, and intellectual condition, and fits to the study the subjects with various backgrounds. The randomness scores in the post-COVID-19 subjects and the normal controls were evaluated in the present study as the behavioral performance, and found no difference between the groups, suggesting that the task performance was not the major factor influencing the results in the Discussion section (line 321).

Methods

5) Subjects: workers in which working field? Workers in the hospital

Response: Thank you for the question. The post-COVID-19 subjects are medical staff. The normal control subjects are not medical staff, as described above, to exclude the subjects with asymptomatic infection from the control group. The details were presented in the Subject subsection of the Material and Methods section (line 116).

6) In what period was the data collected? Please add the information.

Response: Thank you for the comment. The period of the data collection from the post-COVID-19 subjects was between April 2022 and July 2022, and was added to the text (the Materials and Methods section (line 101).

7) Which national or international guidelines on HRV have you used?

Response:  Thank you for the question. The HRV measurement in the present study was based on the standards of the European Society of Cardiology and the North American Society of Pacing and Electrophysiology (Malik, 1996) (line 173).

8) Was participation in the study voluntary?

Response: Thank you for the comment. The subjects voluntarily participated in the present study (line 100).

9) How high was the sample size determined with GPPower?

Response: Thank you for the comment. Sample size was calculated with G-Power and presented in the text when the difference between the data in the post-COVID-19 subjects and that in the normal control subjects was statistically significant (lines 213, 256, 273, 289, 298, 310).

10) Lines 109-111: No statistically significant differences between which groups?

Response: Thank you for the comment. The sentence was changed to ‘No statistically significant differences between the post-COVID-19 and the normal control subjects’ (line 134).

11) HRV analysis should be performed at a high sampling rate (ideally 1,000 Hz)

in order to record the intervals between the individual cardiac actions with high

temporal accuracy. https://register.awmf.org/assets/guidelines/002-042l_S2k_Nutzung-Herzschlagfrequenz-HerzfrequenzvariabilitaetArbeitsmedizin-Arbeitswissenschaft_2022-03_1.pdf

Response: Thank you for the comment. We added the description on our sampling method in the Method section (line 164) as follows, ‘The present ECG device (RF-ECG2, GM3, Tokyo, Japan) is small and wearable to minimize the distress of the subjects, and sampled ECG signals at the frequency of 200 Hz. This sampling rate can cause an inaccurate estimation of R-R interval up to 5 ms, which is 0.5% of the R-R interval data of a subject with the heart rate of 60/min. After peak detection, R-R intervals between the range of 273 and 1500 ms were used for power spectral analysis to exclude paroxysmal beats. When a R-R interval was omitted, it was replaced by the average of the preceding and following intervals. These R-R intervals were resampled at the mean heart rate. This ECG sampling method has been successfully utilized in the previous studies.’

12) For HF and LF, a 5-minute time is recommended for each phase.

https://doi.org/10.3389/fphys.2014.00177

Response: Thank you for the comment. We used MemCalc power spectrum analysis method that can analyze the R-R trend of shorter period instead of FFT. The 5-minute FFT analysis during the task load would cause distress on the subjects and MemCalc has been utilized in the three-behavioral-state paradigm. The details were presented in the Materials and Method section (line 152).

13) Complete the methodological information:

1) reference standard and index

device described in sufficient detail to allow replication (e.g. hardware/software

such as brand, electrode configuration, etc.),

2) reasons for missing data, along with percentage missing (e.g., equipment, persistent ectopy) and how it was handled,

3) interbeat artifact identification method (e.g. algorithm, manual inspection),

4) artifact cleaning methods and percentage of beats corrected. https://doi.org/10.1007/s40279-019-01061-5

Response: Thank you for the comment. The details of the methodological information were added in the Heart rate variability subsection (line 147). The percentage of beats corrected was not available by the present software, but should be small because the subjects with arrhythmic disorders were not enrolled (line 138).

14) Add information on touch performance. What are CB, IB and RNG scores? How were they measured? What are SDS and STAI? How are they measured?

Response: Thank you for the questions. CB is counting bias. IB is interval bias and RNG index is random number generation index. All of them are indices to evaluate the randomness of the generated digit series. Details are found in the text (line 195). SDS and STAI are the questionnaires that are frequently used in the clinical practice to subjectively evaluate the psychophysical conditions (line 123).

15) What about cardiac arrhythmia or extrasystole? Please list separately.

Response:  Thank you for the comment. The subjects with arrhythmic disorders were not enrolled (line 138). And during the measurement, the arrhythmia and extrasystole were excluded from analysis by the software. The details are found in the Materials and Method section (line 164).

16) Data of control group were collected before the outbreak of Covid-19? How can you guarantee equal study conditions? Please describe the examination conditions on the premises?

Response: Thank you for the important comment. We described in detail on this issue in the Introduction section (line 89) and in the Subjects subsection (line 121).

17) What was the physical activity, caffeine intake, tobacco consumption before the examination?

Response: Thank you for the comment. Five minutes of adaptation period without physical activity was introduced before the start of the measurement (line 149). The subject did not take caffein at least an hour before the measurement. Smokers and daytime alcohol drinkers were not enrolled (line 139).

18) Explain random number generation task. What do the test subjects have to do?

Response: Thank you for the comment. In the random number generation task, the subjects generated orally the numbers 0 through 9 at a random order 100 times at the rate of 1Hz. The rate was indicated by a metronome click. The subjects were asked to concentrate on this task, and all completed the task. Randomness in the generated numbers was evaluated using counting bias (CB; frequency of counting-up or down), interval bias (IB; frequency of same interdigit intervals) and random number generation index (RNG; frequency of same digit pairs) according to the previous research (line 195).

19) The Plagscan shows a high level of agreement of 14.4% overall. 13.4% are from Shinba et al. 2023, doi: 10.3390/s23115330. This is most likely due to the methodology. This should be rewritten.

Response: Thank you for the comment. The Heart rate variability measurement subsection in the Materials and Methods section was rewritten (line 145).

Statistics:

20) Is there a normal distribution?

Response: Thank you for the comment. The differences of the HRV and HR indices in each period between the post-COVID-19 and normal control subjects were examined by Man-Whitney U test, because some of the HRV and HR data did not show normal distribution (line 204).

Results:

21) Figure 1: Lines 181-186: already shown under the illustration. This can be deleted from the test.

Response: Thank you for the comment. The descriptions that are shown under the illustration were deleted from the text.

22) The presentation of HRV data in the text is confusing. I suggest creating an additional table with the HRV data and either inserting it or making it available as a supplement.

Response: Thank you for the comment. We made an additional table (Table 1).

23) Lines 300-312: What do previous studies say? One could also controversially discuss that the task did not stress the post-Covid-19 group.

Response: Thank you for the comment. The post-COVID-19 subjects performed the random number generation task with the same randomness level as the normal control subjects revealed by the randomness scores, suggesting that the different stress level during the task load is not the major factor influencing the present results (line 324).

24) On what basis do you argue that post-Covid groups have poor HF adaptation? Can't both groups still be in the range of normal values? There are already various publications with studies on normal values. Please compare your values with normal values from the existing literature.

Response: Thank you for the comment. Following your comment, we modified the discussion. We assessed the change in the autonomic activities at the After state from the initial Rest state, and compared the values in the post-COVID-19 subjects with the values of the normal control (line 318).

25) Lines 325-332: I cannot follow this argument. There is no comparison of the available data with the normal HRV values presented.

Response: Thank you for the comment. We deleted this part from the text.

26) Please introduce a discussion of the limitations. Information on study conditions is currently missing. Different study periods, which can lead to bias. Sampling rate too low with higher source of error, survey period too short (< 5 minutes) per phase (associated with low sampling rate).

Response: Thank you for the comment. Following your comments, a paragraph concerning the limitation was added to the Discussion section (line 360)

Reviewer 2 Report

Comments and Suggestions for Authors

Dear authors, thank you for the intresting study.  just one short recordation , it would be interesting to include in the introduction about communication in the covid for example ( Failure Of Health Diplomacy To Communicate Covid19: Political, Ethical, Legal And Medical Perspective) can be found here:

https://indianmentalhealth.com/pdf/2021/vol9-issue2/13-Ethical-Viewpoint-Paper_Failure-Health.pdf

Comments on the Quality of English Language

Good

Author Response

Dear Reviewer 2

Thank you for your valuable comment.

We modified the manuscript according to your comment. The changes we made are shown below. We hope that the manuscript is improved enough.

Sincerely yours,

Toshikazu Shinba

Department of Psychiatry

Shizuoka Saiseikai General Hospital

Comment 1:

Dear authors, thank you for the interesting study.  just one short recordation, it would be interesting to include in the introduction about communication in the covid for example ( Failure Of Health Diplomacy To Communicate Covid19: Political, Ethical, Legal And Medical Perspective) can be found here:

https://indianmentalhealth.com/pdf/2021/vol9-issue2/13-Ethical-Viewpoint-Paper_Failure-Health.pdf

Response:

Following your comment, we added a description regarding the need for evidence-based clinical communication in the care of COVID-19 infection (Introduction, line 42), and referenced the publication that you introduced us.

Reviewer 3 Report

Comments and Suggestions for Authors

The authors conducted a study of autonomic activity from recovered health workers from mild COVID-19 infection using heart rate variability (HRV). While this study presents compelling insights and potentially valuable data for the field, several issues warrant attention:

1.       Although reference 15 is a peer review chapter from previous work of authors, the affirmations based on this work should be reinforced with other references. Related to lines 86-91, "In healthy subjects, HF reacts to the task load by decrement, and LF/HF by increment. After the task, these changes return to the normal level. In depression, anxiety, and fatigue, not only the Rest data but the Task and After data were altered." Add references from other authors that support the sentence.

2.       Patient description of Methods sections should be integrated into the results sections. This information should be displayed in tables to be easily analyzed. Also, in the Subjects section, please add the COVID vaccination status.  

3.       Incorporating a flow diagram to elucidate the procedural steps involved in the measurements would enhance understanding of the methodology.

4.       For the analysis of the results, using tables in the description of LF, HF, and ratio information would be more valuable. Specifying the scale used on the Y-axis is crucial for accurate interpretation.

5.       The authors should clarify whether the controls are hospital staff members is essential, given the potential association of chronic stress with reduced uncertainty and reduced resting-state heart-rate-variability (HRV) as reported by Jiryis, T., et al. Resting-state heart rate variability (HRV) mediates the association between perceived chronic stress and ambiguity avoidance. Sci Rep 12, 17645 (2022). The discussion of these findings could be especially relevant for health workers.

6.       As most of the subjects analyzed are women (12 males vs 26 females), the authors must discuss if the sexual cycle is relevant in the results, as a recent meta-analysis revealed a significant decrease of HRV in the mid-luteal phase which, as two-follow up studies suggest, is correlated with progesterone. As discussed by Schmalenberger KM, et al. Menstrual Cycle Changes in Vagally-Mediated Heart Rate Variability Are Associated with Progesterone: Evidence from Two Within-Person Studies. J Clin Med. 2020.

7.       The authors must state the hour of the day when the measures were performed for controls and subjects, as LF, HF, and LF/HF follow a circadian pattern. This information is relevant as reported by Deng S, Correlation of Circadian Rhythms of Heart Rate Variability Indices with Stress, Mood, and Sleep Status in Female Medical Workers with Night Shifts. Nat Sci Sleep. 2022

8.       Adding a paragraph explaining how the task was anticipated to induce changes in the autonomic response would elucidate this aspect for readers' comprehension.

9.       Further discussion of task performance in the Results section would provide a deeper understanding of its relevance to the observed outcomes.

10.   Please Add an asterisk as a common way to describe statistical significance; using the connected lines to indicate significant differences could confuse readers. Likewise, the F values displayed do not add more information, nor is a common way to describe differences between groups. In the same way, the authors should describe if normality tests were employed to define the statistical analysis.

Comments on the Quality of English Language

Some typos were found

post-covit19 in line 90

decrease in 182

pari-son in 183

da-ta in 184

Author Response

Dear Reviewer 3

 Thank you for the important and valuable comments and suggestions. We modified the manuscript following them. We hope that the manuscript is improved enough. The modified parts are listed below together with our responses.

Sincerely yours,

Toshikazu Shinba

Department of Psychiatry

Shizuoka Saiseikai General Hospital

General Comment:

The authors conducted a study of autonomic activity from recovered health workers from mild COVID-19 infection using heart rate variability (HRV). While this study presents compelling insights and potentially valuable data for the field, several issues warrant attention:

Comment 1. Although reference 15 is a peer review chapter from previous work of authors, the affirmations based on this work should be reinforced with other references. Related to lines 86-91, "In healthy subjects, HF reacts to the task load by decrement, and LF/HF by increment. After the task, these changes return to the normal level. In depression, anxiety, and fatigue, not only the Rest data but the Task and After data were altered." Add references from other authors that support the sentence.

Response: Thank you for the comment. Following your comment, the original work is cited in the revised manuscript (line 85).

Comment 2. Patient description of Methods sections should be integrated into the results sections. This information should be displayed in tables to be easily analyzed. Also, in the Subjects section, please add the COVID vaccination status. 

Response: Thank you for the comment. We added the vaccination status in the Subjects subsection of the Materials and Methods section (line 106).

Comment 3. Incorporating a flow diagram to elucidate the procedural steps involved in the measurements would enhance understanding of the methodology

Response: Thank you for the comment. We inserted a flow of the paradigm (line 191).

Comment 4. For the analysis of the results, using tables in the description of LF, HF, and ratio information would be more valuable. Specifying the scale used on the Y-axis is crucial for accurate interpretation.

Response: Thank you for the comment. Following your comment, we created a table for the HRV and HRV scores (Table 1).

Comment 5. The authors should clarify whether the controls are hospital staff members is essential, given the potential association of chronic stress with reduced uncertainty and reduced resting-state heart-rate-variability (HRV) as reported by Jiryis, T., et al. Resting-state heart rate variability (HRV) mediates the association between perceived chronic stress and ambiguity avoidance. Sci Rep 12, 17645 (2022). The discussion of these findings could be especially relevant for health workers.

Response: Thank you for the important comment. We used the normal controls subjects before the outbreak of COVID-19 infection and were not the medical staff because it was not possible to enroll the medical staff completely free from the COVID-19 infection. As you pointed out that the medical environment with chronic stress may induce reduction of HRV. However, our data indicated that the resting HRV scores were not altered in the post-COVID-19 subjects. Further, the psychophysical conditions were checked in both groups using questionnaires and the questionnaire scores in two groups are not different (line 116). The paradigm and methods we employed in the present study were also completely same as those used for the normal control subjects. Based on these backgrounds, we discussed the HRV and HR data in the post-COVID-19 subjects and those in the control subjects. We also described the limitation of the present finding in the Discussion section (line 360) by referencing the literature you recommended.

Comment 6. As most of the subjects analyzed are women (12 males vs 26 females), the authors must discuss if the sexual cycle is relevant in the results, as a recent meta-analysis revealed a significant decrease of HRV in the mid-luteal phase which, as two-follow up studies suggest, is correlated with progesterone. As discussed by Schmalenberger KM, et al. Menstrual Cycle Changes in Vagally-Mediated Heart Rate Variability Are Associated with Progesterone: Evidence from Two Within-Person Studies. J Clin Med. 2020.

Response: Thank you for the important comment. The menstrual cycle of the female subjects was not available in the present study. The need to analyze it in the future studies was described in the limitation paragraph in the Discussion section (line 373). However, the HRV and HR scores in the post-COVID-19 subjects were not altered at the resting state, suggesting that the hormonal influence was not the major factor influencing the present results (line 367).

Comment 7. The authors must state the hour of the day when the measures were performed for controls and subjects, as LF, HF, and LF/HF follow a circadian pattern. This information is relevant as reported by Deng S, Correlation of Circadian Rhythms of Heart Rate Variability Indices with Stress, Mood, and Sleep Status in Female Medical Workers with Night Shifts. Nat Sci Sleep. 2022

Response: Thank you for the important comment. HRV measurement was conducted between 14:00 and 17:00 (line 149). The detailed analysis on the circadian profiles of HRV was not possible in the present study. Future studies regarding this issue would be interesting. This description was added to the limitation paragraph in the Discussion section (line 369).

Comment 8. Adding a paragraph explaining how the task was anticipated to induce changes in the autonomic response would elucidate this aspect for readers' comprehension.

Response: Thank you for the comment. The task is anticipated to induce changes as follows in the healthy subjects; ‘In healthy subjects, HF reacts to the task load by decrement, and LF/HF by increment. After the task, these changes return to the normal level’. This description was presented in the Introduction section (line 85).

Comment 9. Further discussion of task performance in the Results section would provide a deeper understanding of its relevance to the observed outcomes.

Response: Thank you for the comment. We added the discussion on significance of task performance in relation to the HRV and HR data in the Discussion section (line 321).

Comment 10. Please Add an asterisk as a common way to describe statistical significance; using the connected lines to indicate significant differences could confuse readers. Likewise, the F values displayed do not add more information, nor is a common way to describe differences between groups. In the same way, the authors should describe if normality tests were employed to define the statistical analysis.

Response: Following your comments, we added an asterisk to describe the statistical significance in Figure 2. F values were deleted from the Figures. Normality tests revealed that some data in the present studies did not show normal distribution and we used Mann Whitney U test for analyzing the difference in two groups (line 207).

Comments on the Quality of English Language

Some typos were found

post-covit19 in line 90

decrease in 182

pari-son in 183

da-ta in 184

Response: Thank you for the comments. We corrected the errors.

Round 2

Reviewer 1 Report

Comments and Suggestions for Authors

Thank you for processing and considering my comments. All the best for your future research. 

Comments on the Quality of English Language

As a non-native English speaker, the publication reads well. 

Author Response

Dear Reviewer 1

Thank you very much for your previous comments and suggestions, and for your help in improving our manuscript.

Comment 1

Thank you for processing and considering my comments. All the best for your future research.

Response: We are glad to know that our responses are well enough.

Comment 2

As a non-native English speaker, the publication reads well.

Response: Thank you for checking the language in our manuscript.

We thank you very much again for helping us to improve the manuscript.

Sincerely yours,

Toshikazu Shinba

Department of Psychiatry

Shizuoka Saiseikai General Hospital